# Light- and Redox-Responsive Block Copolymers of mPEG-SS-ONBMA as a Smart Drug Delivery Carrier for Cancer Therapy

**DOI:** 10.3390/pharmaceutics14122594

**Published:** 2022-11-24

**Authors:** Yu-Lun Lo, Yao-Hsing Fang, Yen-Ju Chiu, Chia-Yu Chang, Chih-Hsien Lee, Zi-Xian Liao, Li-Fang Wang

**Affiliations:** 1Department of Medicinal and Applied Chemistry, College of Life Sciences, Kaohsiung Medical University, Kaohsiung 807, Taiwan; 2Institute of Medical Science and Technology, National Sun Yat-Sen University, Kaohsiung 804, Taiwan; 3Department of Medical Research, Kaohsiung Medical University Hospital, Kaohsiung 807, Taiwan

**Keywords:** micelle, *o*-nitrobenzyl methacrylate, dual-stimuli response, atom transfer radical polymerization, drug delivery system

## Abstract

The development of stimuli-responsive polymeric micelles for targeted drug delivery has attracted much research interest in improving therapeutic outcomes. This study designs copolymers responsive to ultraviolet (UV) light and glutathione (GSH). A disulfide linkage is positioned between a hydrophilic poly(ethylene glycol) monomethyl ether (mPEG) and a hydrophobic *o*-nitrobenzyl methacrylate (ONBMA) to yield amphiphilic copolymers termed mPEG-SS-pONBMA. Three copolymers with different ONBMA lengths are synthesized and formulated into micelles. An increase in particle size and a decrease in critical micelle concentration go together with increasing ONBMA lengths. The ONB cleavage from mPEG-SS-pONBMA-formed micelles results in the transformation of hydrophobic cores into hydrophilic ones, accelerating drug release from the micelles. Obvious changes in morphology and molecular weight of micelles upon combinational treatments account for the dual-stimuli responsive property. Enhancement of a cell-killing effect is clearly observed in doxorubicin (DOX)-loaded micelles containing disulfide bonds compared with those containing dicarbon bonds upon UV light irradiation. Collectedly, the dual-stimuli-responsive mPEG-SS-pONBMA micelle is a better drug delivery carrier than the single-stimuli-responsive mPEG-CC-pONBMA micelle. After HT1080 cells were treated with the DOX-loaded micelles, the high expression levels of RIP-1 and MLKL indicate that the mechanism involved in cell death is mainly via the DOX-induced necroptosis pathway.

## 1. Introduction

Polymeric micelles of different shapes and sizes are crucial to cancer diagnosis and therapy because of their promising advantages, including increased drug accumulation at the tumor site and reduced side effects under the enhanced permeability and retention (EPR) effect [1,2]. The hydrophobic cores of polymeric micelles can be utilized for loading hydrophobic drugs and regulating drug release behaviors, while hydrophilic shells increase the solubility of hydrophobic drugs [3]. However, self-assembled micelles can be disrupted upon large dilution in the bloodstream and dissociate at a concentration below the critical micelle concentration (CMC). This situation accelerates premature drug release at normal tissues or organs, leading to reducing drug accumulation at the target site [4]. Preparing self-assembled micelles from amphiphilic block copolymers with a low CMC value for drug delivery is of need. Furthermore, the design of amphiphilic block copolymers sensitive to light is attractive owing to that light can direct active molecules into a target site in high spatiotemporal precision with various selection wavelengths [5]. Recently, self-assembled micelles with light-cleavable *o*-nitrobenzyl (ONB) moiety are widely applied to drug delivery systems (DDS) [6,7,8]. A well-reviewed article has documented several examples of applying ONB esters for the design of photo-responsive polymer network [9].

The ONB moiety can be introduced into amphiphilic block copolymers either as pendent groups or in the backbone chain of parent block copolymers. In our previous study, we synthesized an amphiphilic block copolymer with adjacent disulfide bonds and ONB moieties for dual stimuli-responsive DDS [10]. However, the amphiphilic block copolymer with a single-stimulus ONB group positioned at the backbone may require a higher concentration or longer irradiation time to achieve micellar degradation and accelerate drug release at the target site. Herein, a methacrylate monomer containing ONB (ONBMA) was synthesized and proceeded to form an amphiphilic block copolymer comprising a hydrophilic poly(ethylene glycol) (PEG) block and a hydrophobic ONBMA block. Responsiveness to ultraviolet (UV) light can be easily tuned by controlling the numbers of ONBMA in the block copolymers. Dong et al. have synthesized poly(ethylene glycol)-SS-[poly(dimethylaminoethyl methacrylate)-co-poly(2-nitrobenzyl methacrylate)] [PEG-SS-(PDMAEMA-co-PNBM)] and demonstrated a quadruple responsive property of the copolymer; nevertheless, the authors did not study any potentials of using this copolymer for DDS [11].

In this study, novel dual-stimuli cleavable amphiphilic block copolymers containing a dual response to UV light and glutathione (GSH) are synthesized to accelerate both the release and accumulation of chemo drugs at the tumor site. The amphiphilic block copolymers are synthesized with a hydrophilic PEG segment and a hydrophobic ONBMA segment via atom transfer radical polymerization (ATRP). The PEG segment is functionalized with bromide and disulfide moieties for proceeding to the ATRP reaction and endowing a redox-responsive property. ATRP is one of the living radical polymerization techniques widely applied to DDS [12,13,14,15,16,17]. The PEG-based macro initiators with either a disulfide linkage (-SS-) or a dicarbon linkage (-CC-) are synthesized and positioned in the hydrophobic and hydrophilic junction of the amphiphilic block copolymer. The hydrophobic ONBMA segment of the amphiphilic copolymer undergoes photolysis upon UV light irradiation, resulting in the transformation of the hydrophobic ONBMA segment to the hydrophilic MAA segment. The disulfide unit is degraded in the presence of a reducing agent such as dithiothreitol (DTT) or GSH. GSH is an endogenous reducing agent with various concentrations in intracellular (1–10 mM) and extracellular areas (2–20 μM) of living cells [18]. In addition, the GSH concentration in cancer cells is several times higher than that in normal cells [19]. Thus, preparing a drug carrier responsive to a GSH concentration gradient is an attractive cue to triggering drug release at the tumor site. These artificially designed amphiphilic block copolymers endow their micellar degradation and drug release via three mechanisms, namely, hydrophobic–hydrophilic transformation, photocleavage of ONB bonds, and reductive cleavage of disulfide bonds.

An amphiphilic block copolymer without reductive cleavage of dicarbon bonds is synthesized for comparison as well. A model drug, doxorubicin (DOX), is encapsulated into the hydrophobic core of both micelles. The DOX-loaded micelles with disulfide linkages are thoroughly characterized to demonstrate a higher anticancer effect than those with dicarbon linkages. The mechanism of DOX-induced cell death is also well-assayed.

## 2. Materials and Methods

### 2.1. Materials

2-Bromo-2-methylpropionyl bromide, triethylamine (TEA), N,N′-dicyclohexylcarbodiimide (DCC), 4-dimethylaminopyridine (DMAP), and copper(I) chloride (CuCl) were purchased from Alfa Aesar (Heysham, UK). Polyethylene glycol monomethyl ether (mPEG, Mw = 2000 g/mol) was obtained from TCI (Tokyo, Japan). N,N,N′,N″,N″-Pentamethyldiethylenetriamine (PMDETA), 2-hydroxyethyl disulfide, o-nitrobenzyl alcohol and succinic anhydride were purchased from Sigma-Aldrich (St. Louis, MO, USA). 3-(4,5-Dimethylthiazol-2-yl)-2,5-diphenyl-tetrazolium bromide (MTT) was from MP Biomedicals (Santa Ana, CA, USA). Phosphate buffer saline (PBS), Dulbecco’s Modified Eagle Medium (DMEM), Roswell Park Memorial Institute (RPMI) 1640 Medium, trypsin–EDTA, and fetal bovine serum (FBS) were purchased from Invitrogen (Waltham, MA, USA). Doxorubicin hydrochloride (DOX·HCl) was purchased from Combi-Blocks (San Diego, CA, USA).

### 2.2. Synthesis of mPEG-SS-Br Macroinitiator (***3***)

2-Hydroxyethyl-2′-(bromoisobutyryl) ethyl disulfide (HO-SS-Br, **1**) was synthesized according to previous publication [20]. The final product was purified by column chromatography with a 60–200 um silica column, and the eluent is 20% *v/v* ethyl acetate in hexane. The yield is ~35.5%. ^1^H-NMR (Bruker AM 400, CDCl_3_): δ (ppm) 4.45 (t, 2H), δ 3.90 (t, 2H), δ 2.98 (m, 2H), δ 2.90 (m, 2H), δ 1.95 (s, 6H).

mPEG-COOH (**2**) was synthesized by esterification reaction of methoxy end-capped mPEG and succinic anhydride according to the literature [21]. After three times precipitation using DCM as a solvent and ethyl ether as a non-solvent, the final precipitate was filtered and dried under vacuum with a 90.0% yield. ^1^H-NMR (Bruker AM 400, CDCl_3_): δ (ppm) 4.20 (m, 2H), δ 3.33 (s, 3H), δ 2.58 (m, 4H).

mPEG-SS-Br macroinitiator (**3**) was synthesized according to the literature with slight modification [21]. Briefly, HO-SS-Br (**1**) (406 mg, 1.3 mol) was put into a 50 mL two-neck bottle protected with argon, followed by sequentially adding 20 mL anhydrous dichloromethane (DCM), DMAP (41 mg, 0.3 mmol) and DCC (1.04 g, 5 mmol). The reaction proceeded in the bottle immersed in an ice bath for 2 h, which was then moved to an oil bath at 35 °C, followed by addition of mPEG-COOH (**2**) (720 mg, 0.3 mmol) with continuous stirring for 48 h under argon. After reaction, insoluble dicyclohexylurea byproduct was filtered and the solution was precipitated into hexane. The precipitate was collected and dissolved in DCM, followed by precipitation into hexane again. The solid was collected and dissolved in DCM and precipitated into ethyl ether to remove excess HO-SS-Br. The final product was filtered and dried in a vacuum oven for 24 h at room temperature (RT) with a 94.2% yield. ^1^H-NMR (Bruker AM 400, CDCl_3_): δ (ppm) 4.16 (m, 2H), δ 3.95 (m, 4H), δ 3.32 (s, 3H), 2.90 (m, 4H), 2.60 (m, 4H), 1.93(s, 6H).

### 2.3. Synthesis of ONBMA (***4***)

o-Nitrobenzyl methacrylate (**4**, ONBMA) was prepared using a procedure slightly modified from the literature [22]. Briefly, *o*-nitrobenzyl alcohol (5 g, 32 mmol) was dissolved in anhydrous DCM (100 mL) in a 250 mL flask and protected with argon for 30 min. TEA (5 mL, 35 mmol) was added into the above solution in an ice/water bath, followed by adding methacryloyl chloride (5.7 g, 58 mmol) dropwise. After 24 h, the solution was extracted sequentially with an aqueous solution of NaHCO_3_ (100 mL, 10 wt%), 1N HCl, NaCl saturated liquid, and double deionized (DD) water. The DCM layer was separated and dried. The crude product was purified using silica gel chromatography with 25:0.2 hexane/ethyl ether. The pure product was obtained as a light yellowish liquid (yield: 36%). ^1^H-NMR (Bruker AM 400, CDCl_3_): δ (ppm) 8.15 (m, 1H), δ 7.50 (m, 3H), δ 6.15 (s, 1H), δ 5.52(s, 1H), δ 5.5 (s, 2H), δ 1.92 (s, 3H).

### 2.4. Synthesis of mPEG-SS-pONBMA (***5***)

The amphiphilic block copolymer of mPEG-SS-pONBMA (**5**) was synthesized via ATRP. Macroinitiator mPEG-SS-Br (100 mg, 0.04 mmol) and different concentrations of ONBMA were put in a 5 mL flask degassed under vacuum at 30 °C to remove moisture, followed by addition of anhydrous dimethylformamide (DMF, 3 mL) using a syringe. The above solution was degassed three times using liquid nitrogen and sealed under vacuum. The degassed flask was then moved into a glove box filled with nitrogen. PMDETA (4.3 μL, 0.02 mmol) and CuCl (2 mg, 0.02 mmol) were added into the flask, and the ATRP reaction was carried out at 80 °C for 15 h. After reaction, the solution was cooled to RT and poured into ethyl ether at 0 °C. The precipitate was collected and dissolved in DCM, followed by precipitation into hexane to remove ONBMA. The precipitated product was collected and dissolved in DCM and precipitated again in ethyl ether to remove DMF residue, and a light blue powder was obtained and dried in a vacuum oven for 24 h at RT. This powder was dissolved in THF and passed through a neutral Al_2_O_3_ column to remove the catalyst. The pure product was yielded as a light yellowish powder.

### 2.5. Characterization of Amphiphilic Block Copolymers

^1^H-nuclear magnetic resonance (^1^H-NMR) spectra were recorded on a Bruker AM 400 (Billerica, MA, USA) NMR spectrometer. Size exclusion chromatography (SEC) was acquired using Agilent (Santa Clara, CA, USA) 1200 series equipped with Shodex KF-804L and KF-803 connected columns and a refractive index detector. THE was used as an eluent at a flow rate of 1 mL/min at 30 °C. The Fourier transform infrared (FTIR) spectrum was acquired from the Bruker ALPHA-T spectrometer. Samples were ground with the KBr pellet and pressed into a thin film. Mass data were acquired from the TRACEGC-POLARISQ (Thermo/Finnigan, San Jose, CA, USA) mass spectrometer.

### 2.6. Preparation and Characterization of Micelles

#### 2.6.1. Micelle Formation

A nanoprecipitation method [23] was acquired for formulating micelles with a concentration of a copolymer at 5 mg/mL in THF, dropwise added into 50 mL DD water using a syringe pump within 30 min. THF was removed using Rotavapor under reduced pressure. The micelle formulated from mPEG-SS-pONBMA is abbreviated as MSP and that from mPEG-CC-pONBMA as MCP. The number indicated in the abbreviation means the repeating unit of ONBMA. The morphologies of micelles were observed using a transmission electron microscope (TEM, HT7700, Hitachi, Tokyo, Japan), and the size distribution was determined using dynamic light scattering (DLS, ELSZ-2000, Otsuka Electronics, Osaka, Japan). The critical micellar concentrations (CMC) of micelles were determined using a Cary Eclipse fluorescence spectrophotometer (Varian, Palo Alto, CA, USA) with pyrene as a probe. The details of TEM, DLS, and CMC measurements can be referred to in our previous publication [24].

#### 2.6.2. Reductive- and Photo-Triggered Cleavage of Micelles

Redox-stimuli cleavage of micelles was carried out in 5 mM GSH solution for 20 min. Micelles were dispersed in DD water containing 0.5 mg/mL GSH. For UV-triggered cleavage, 2 mL micellar aqueous solution was placed into a cuvette and stirred continuously. The cuvette was kept in the dark and irradiated by continuous wave (CW) UV laser diode (365 nm, max. 430 mW/cm^2^) horizontally under continuous stirring for 20 min. For dual-triggered cleavage of the micelle, the concentration of GSH was reduced to 2.5 mM and the UV irradiation time was cut to 10 min. The size and morphology changes in micelles after treatments were observed using DLS and TEM.

### 2.7. DOX-Encapsulated Micelles and DOX Release

The DOX∙HCl powder was dissolved in dimethyl sulfoxide (DMSO) at a concentration of 2 mg/mL and an equimolar amount of TEA was added for desalting. The DOX solution (1 mL) was added into the solution containing 20 mg of PEG-SS-pONBMA or PEG-CC-pONBMA in DMSO according to our previous publication [10]. A dried sample was dissolved in DMSO at a concentration of 1 mg/mL, and the DOX content was calculated against a calibration curve using a fluorescence spectrometer with the emission intensity at 585 nm. The encapsulation efficiency (EE) and loading efficiency (LE) were reported as follows.
EE (%) = (amount of DOX in micelle/amount of DOX in feed) × 100(1)
LE (%) = (amount of DOX in micelle/total amount of micelle and DOX) × 100(2)

The external-trigger-released behavior of DOX from micelles was determined in PBST (PBS + 0.2% Tween20) at pH 7.4 and 37 °C for simulating a physiological condition. Each 800 μL was withdrawn from DOX-loaded micellar solution at 3 mg/mL in PBST and put into a DiaEasy™ Dialyzer (MWCO 3.5 kDa, BioVision, Milpitas, CA, USA) and immersed in 12 mL of PBST under four different conditions: (1) without any treatment (control), (2) in the presence of 5 mM GSH, (3) under UV irradiation for 7 min, (4) in the presence of 2.5 mM GSH and under UV irradiation for 3.5 min. At a certain time, 3 mL of PBST solution was carefully removed from the tube and replaced with 3 mL of fresh PBST to maintain a sink condition. The DOX concentration was determined as aforementioned.

### 2.8. Cell Experiments

#### 2.8.1. Cell Culture

Non-small cell lung carcinoma A549 cells were purchased from ATCC (Manassas, VA, USA) and cultured in DMEM, and fibrosarcoma HT1080 cells were purchased from Bioresource Collection Research Centre (BCRC, Hsinchu, Taiwan) and cultured in MEM, respectively, containing 10% fetal bovine serum (FBS) and 100 μg/mL penicillin/streptomycin in a 37 °C incubator with a humidified atmosphere containing 5% CO_2_.

#### 2.8.2. Relative Cell Viability of Micelles and DOX-loaded Micelles

Cell viability was determined with an MTT assay against A549 cells and H1080 cells. To assay the photo-triggered cytotoxicity of DOX-loaded micelles, cells were seeded into 96 well plates (5 × 10^3^ cells/well) for 24 h and added with various concentrations of test samples and post incubated for another 48 h. The cytotoxic efficiency of DOX-loaded micelles and free DOX was determined with the MTT assay [19].

#### 2.8.3. Cellular Uptake and Apoptosis

The intracellular uptake of DOX-loaded micelles was observed with a confocal laser scanning microscope (CLSM, LSM 700 Zeiss Confocal Microscopy). HT1080 cells (5 × 10^4^ cells/well) were seeded in 4-well chambers and incubated for 24 h. The cells were treated with free DOX or DOX-loaded micelles at 1 μg/mL. Following 4 h incubation, the medium was replaced with 1 mL of fresh complete medium, and the cells were exposed to UV light (365 nm, 430 mW/cm^2^) for 10 min and postincubated for another 2 h. The Annexin-V/PI (propidium iodide) dual staining assay was performed to estimate apoptosis-inducing efficacy of free DOX, and DOX-loaded micelles before and after UV light irradiation as well.

#### 2.8.4. Western Blot Analysis

HT1080 cells were homogenized in ice-cold radioimmunoprecipitation assay (RIPA) buffer. The total protein concentration was determined by Bicinchoninic Acid (BCA) Protein Assay kit (Thermo), according to the manufacturer’s instructions. Samples containing 10 μg proteins were separated via sodium dodecyl sulfate–polyacrylamide gel electrophoresis (SDS-PAGE), and the proteins were transferred to polyvinylidene difluoride (PVDF) membrane (Millipore, Bedford, MA, USA). After blocking with a 5% skimmed milk for 1 h at RT, the membranes were incubated overnight at 4 °C with the following primary antibodies: anti-RIP1 monoclonal antibody (Cell Signaling Technologies, Danvers, MA, USA), anti-MLKL monoclonal antibody (Cell Signaling Technologies, Danvers, MA) and anti-GAPDH polyclonal antibody (GeneTex International Corporation, Hsinchu, Taiwan). Protein expression was quantified via Image J. The quantitative values for each protein were normalized to control group.

### 2.9. Statistical Analysis

Experiments were repeated at least three times, and data are expressed as the mean ± standard deviation. Student’s t-test was used to determine the statistical significance of the respective group. * *p* < 0.05, ** *p* < 0.005 and *** *p* < 0.0005 indicate a significant difference.

## 3. Results and Discussion

### 3.1. Copolymer Synthesis and Characterization

Figure 1 illustrates the synthesis route of amphiphilic mPEG-SS-ONBMA copolymers. Initially, Compound **1** (HO-SS-Br) was prepared according to a reported procedure [20], and its ^1^H-NMR and LC/MS spectra are consistent with the literature (Appendix A). Secondly, Compound **2** (mPEG-COOH) was synthesized by an esterification reaction between methoxy end-capped PEG and succinic anhydride [21]. The synthesis of Compound **3** (mPEG-SS-Br) was carried out through the condensation of Compounds **1** and **2** in the presence of DCC and DMAP (Appendix A) [21]. The FTIR spectrum of mPEG-COOH revealed the clear characteristic peak of the carboxylic C=O stretching band at 1734 cm^−1^, while mPEG-SS-Br exhibited at 1736 cm^−1^ (Appendix A). The number-averaged molecular mass of mPEG-SS-Br was 2400 g/mol with a dispersity of 1.07 measured by SEC. Thirdly, Compound **4** (ONBMA) was synthesized, using a procedure slightly modified from the literature [22], as a light yellowish liquid with a yield of 36% (Appendix A).

Three mPEG-SS-ONBMA copolymers with different hydrophobic ONBMA lengths were prepared using a fixed amount of mPEG-SS-Br (0.04 mmol) but various amounts of ONBMA (2, 4, and 8 mmol). Figure 1A shows the NMR spectrum of the copolymer with 8 mmol ONBMA infeed. The proton intensity ratio of peaks attributed to three aromatic protons at ~8.0 ppm (peak J) and those protons attributed to (-CH_2_CH_2_O-)_45_ of mPEG at ~3.3 ppm were adopted to calculate the degree of polymerization of ONBMA, i.e., ~18, 40, and 50, and copolymers were thus abbreviated as MSP_18_, MSP_40_, and MSP_50_, respectively. Table 1 lists the number-averaged molecular weights of the copolymers calculated using SEC profiles (Figure 1C). The molar mass increased with increasing ONBMA length from 2900 g/mol for MSP_18_ to 7800 g/mol for MSP_50_.

### 3.2. Formulation and Characterization of Micelles

Micelles were prepared through nanoprecipitation [23]. The hydrodynamic particle diameters of MSP_18_, MSP_40_, and MSP_50_ micelles were ~95 nm, 134 nm, and 204 nm, and their corresponding TEM images were included in Appendix A. The responsiveness of three micelles to GSH and UV light was preliminarily tested with different treatments, i.e., 5 mM GSH for 20 min, a UV light for 20 min, and the combination of 2.5 mM GSH and UV light for 10 min. According to DLS results, MSP_18_ was irresponsive to a single trigger but fast responsive to the combinational triggers of GSH and UV light. Nevertheless, MSP_40_ and MSP_50_ showed good responsiveness to both the GSH and UV light (Appendix A). The micelles were stable at 0.1 mg/mL in an aqueous solution for 7 days (Appendix A).

CMC values were in the range of 1.95 × 10^−4^–1.22 × 10^−5^ mg/mL and increased with increasing lengths of ONBMA assayed with a pyrene probe (Appendix A, Table 1). Because such low CMC values are seldom observed, we prepared a series concentration gradient within 6.0 × 10^−6^–0.01 mg/mL of amphiphilic copolymers to double-check the CMC values using a DLS technique. Indeed, self-assembled particles were formed in the concentration within 10^−5^–10^−4^ mg/mL (Appendix A, Table 1**)**. Compared with our previously synthesized copolymer [10] and the one from the literature [25] with the ONB moiety positioned in the main chain, MSPs listed in Table 1 had extremely lower CMC values (10^−4^–10^−5^ mg/mL) versus (10^−1^–10^−3^ mg/mL) [10,25]. The CMC values were also much lower than those of amphiphilic random copolymers where the ONB moiety was positioned in the side chain. They were 0.05 mg/mL for poly(*o*-nitrobenzyl methacrylate)-*co*-2-(2-methoxyethoxy)ethyl methacrylate -*co*-oligo(ethylene glycol) methacrylate [P(ONBMA-*co*-MEO_2_MA-*co*-OEGMA)] [26], 0.12 mg/mL for polyhedral oligomeric silsesquioxane end-capped poly(*o*-nitrobenzyl methacrylate) (POSS–ONBMA) [27] and 0.15 mg/mL for [PEG-SS-(PDMAEMA-co-PNBM)], which owned a similar structure to this study [11]. The micelle with a low CMC value was beneficial for DDS owing to both increased dilution stability and inhibited drug release under circulation in the bloodstream [28].

With the largest hydrophobic capacity of MSP_50_ to accommodate the largest amount of hydrophobic anticancer drugs, a comparable copolymer without GSH responsiveness, mPEG-CC-pONBMA (MCP_50_), was prepared using a similar synthesis route as indicated in Figure 1, except that the disulfide bonds of Compound **1** were replaced with dicarbon bonds. Figure 1B displays the NMR spectrum of the MCP_50_ copolymer with the number of ONBMA repeating units close to that of MSP_50_. The CMC values of both micelles were analogous. The hydrodynamic diameter of the MCP_50_ micelle (125 nm) was smaller than that of the MSP_50_ micelle (204 nm). This phenomenon might be attributed to the stronger hydrophobic interaction of -CC- linkages than those of -SS- linkages [29].

### 3.3. Stimuli-Responsive Properties of MSP_50_ and MCP_50_

We continued to work on MSP_50_ and MCP_50_ micelles because their equivalent numbers of ONBMA block might show similar responsiveness to UV light. To evaluate the responsivity of MSP_50_ and MCP_50_ to the physiological condition, copolymers and their formulated micelles were treated with 5 mM GSH for various durations. The DLS profiles of the MSP_50_ and MCP_50_ micelles showed a single and sharp distribution in the absence of GSH, changed to bimodal distribution upon 1 min GSH treatment, and shifted to larger particle size distribution with prolonged time incubation (Figure 2A,C). The SEC profiles of the MCP_50_ copolymer remained intact with time (Figure 2B) but those of MSP_50_ showed random distribution and shifted to lower MW upon 30 min GSH treatment (Figure 2D), explaining the degradation of disulfide bonds with GSH. Furthermore, the SEC profiles of the copolymers and the DLS diagrams of the micelles were traced with different treatments. As shown in Figure 3A, the same result was found as aforementioned in MCP_50_ treated with 5 mM GSH for 20 min, i.e., the dramatic increase in particle size was associated with the invisible change in molecular weight by SEC. To account for this phenomenon, the formation of hydrogen bonding between MCP_50_ and GSH was proposed in Figure 2A and evidenced by FTIR. Comparison with the FTIR spectrum of the nascent MCP_50_ revealed a broader -OH stretching peak around 3410 cm^−1^ and the C=O stretching peak shifted from 1730 cm^−1^ to 1710 cm^−1^, thus verifying the H-bonding formation in the presence of GSH (Appendix A). Accordingly, the hydrogen bonding interaction should exist between MSP_50_ and GSH; however, the simultaneous cleavage of disulfide bonds leads to the disassembly of micelles and results in the indistinct observation of particle aggregation.

Both particle size and molecular weight distribution of MCP_50_ changed obviously upon UV light stimulation (Figure 3A). On the other hand, Figure 3B shows that MSP_50_ had a better response to GSH and UV light and synergistically enhanced degradation even at half the amount of both stimuli. The morphological changes in MCP_50_ and MSP_50_ were also traced by TEM. The spherical particle shape observed in MSP_50_ (~150 nm) became irregular aggregation when treated with 10 mM GSH or UV light irradiation for 20 min and changed into fragments when treated with both stimuli in combination at half their amounts (Figure 3C), indicating that both the disulfide linkage and ONB moiety were degradable. In contrast, severe particle aggregation without degradation was clearly observed in the MCP_50_ micelle treated with GSH. Fuzzy particulate morphology was seen in the MCP_50_ micelle treated with UV light. Compared with that treated with 10 mM GSH, the MCP_50_ micelle treated with 5 mM GSH, and UV light showed less particle aggregation (Figure 3C), implying that the degree of H-bonding formation enhanced with increasing GSH concentration.

Figure 2B shows light- and redox-stimuli responsiveness of MSP. Under UV light absorption, the nitro groups located at the *ortho* position of the benzene ring caused degradation of ester bonds, resulting in the release of *o*-nitrosobenzaldehyde groups from ONBMA. The signal peak of the aldehyde fragment from the MSP copolymer appearing at 10.2 ppm became clearer in the NMR spectrum as increasing UV irradiation time from 10 to 20 min (Appendix A). The light- and redox-responsive properties of MSP have been evidenced by SEC (Figure 3B). After GSH treatment, the molecular weight distribution of MSP shifted from high to low distribution. Nevertheless, after UV light treatment, the removal of *o*-nitrosobenzaldehyde groups from either MSP or MCP led to increased water solubility. This fact hampered the MW trace by SEC. Alternatively, the UV–vis spectra of MSP and MCP dispersed in an aqueous solution were simultaneously monitored to confirm photocleavage of *ONB* upon UV light irradiation. A decrease in peak intensity at ~ 270 nm, attributable to the absorption of *o*-nitrobenzyl esters, was associated with an increase in peak intensity at ~320 nm corresponding to the absorption of labile *o*-nitrosobenzaldehyde groups (Appendix A). The degradation kinetics of ONB from MSP and MCP was the same because a similar release profile of the *o*-nitrosobenzaldehyde group was followed with time under UV light irradiation (Appendix A). In addition, the solution of MSP and MCP gradually changed from colorless to slightly yellowish upon UV light irradiation.

### 3.4. Drug Encapsulation and Cytotoxicity

The desalted DOX molecule was used as a drug model. DOX is an anthracycline antibiotic, widely used to treat solid tumors such as lung, breast, ovarian, thyroid, and gastric cancers [30]. The encapsulation and loading efficiencies of DOX were 56%, ~5.0% for DOX-loaded MSP micelle (MSP-D) and 70%, and 6.7% for DOX-loaded MCP micelle (MCP-D), respectively. In vitro release of DOX from the micelle was performed in PBST solution at pH 7.4 under four different conditions. Figure 4 shows the gradual release of DOX from MCP-D and MSP-D with time. The amount of DOX released from MSP-D was consistently larger than that from MCP-D in all four test conditions. About 100% of DOX was released from MSP-D within 12 h, compared with ~40% from MCP-D with 5 mM GSH. GSH concentrations in intracellular areas (1–10 mM) were much higher than those in extracellular areas (2–20 μM) in living cells [18]; moreover, those concentrations in cancer cells were also several times higher than those in normal cells [19]. Thus, the responsive degradation of amphiphilic block copolymers with GSH concentration gradient is an attractive approach to eliciting drug release to the tumor microenvironment (TME) [21,31,32]. A drug release profile of MSP-D treated with the dual stimuli of 3.5 min UV light and 2.5 mM GSH was like that treated with 7 min UV light. The use of shorter irradiation time to trigger drug release was preferred and beneficial because UV light was considered to be carcinogenic and could induce tissue damage [5].

The cytotoxicity of MCP_50_ and MSP_50_ was tested against A549 cells and HT1080 cells (Figure 5A). For 24 h post-incubation, the micelle showed no obvious cytotoxicity at the concentration < 100 μg/mL in both the cells, having ~100% relative cell viabilities. However, slight cytotoxicity was found in MCP_50_ against A549 cells (~76%) and MSP_50_ against HT1080 cells (~82%) when the concentration was increased to 200 μg/mL. Figure 5B,C shows the relative cell viabilities of A549 cells and HT1080 cells exposed to free DOX, MCP-D, and MSP-D at various equivalent DOX concentrations for 48 h post-incubation. The IC_50_ values calculated according to the concentration of DOX required to inhibit 50% of cell proliferation were 1.71 μM for free DOX, 1.69 μM for MCP-D, and 1.25 μM for MSP-D against HT1080 cells, and 8.15 μM, 3.33 μM and 4.16 μM, respectively, against A549 cells. The superior apoptotic potency over free DOX was clearly observed in MSP-D. The IC_50_ values of HT1080 cells were lower than those of A549 cells. This fact may be because HT1080 was a newly purchased cell line but a repeated treatment of A549 cells with DOX-induced resistance to chemotherapy and development of multidrug resistance [33]. Thus, the cellular behavior of DOX-loaded micelles was focused on HT1080 cells alone.

The cellular uptake of DOX was traced by CLSM. Figure 6A shows the highest fluorescence intensity of HT1080 cells exposed to MSP-D under UV light irradiation, indicating an enhanced DOX release by cleavage of ONB linkages and disulfide bonds. The red fluorescence of CLSM images was clearly observed in cells exposed to free DOX with no visible difference between cells treated with and without UV light.

Annexin-V/PI dual-staining assay was acquired for apoptosis analysis and categorized into four quadrants: the upper left quadrant (Q1) indicates necrotic cells stained with PI; the upper right quadrant (Q2) indicates late apoptotic cells stained with PI and Annexin-V; the lower right quadrant (Q3) indicates early apoptotic cells stained with Annexin-V, and the lower left quadrant (Q4) indicates healthy cells not stained with PI and Annexin-V [34]. The percentages of Annexin-V/PI staining positive cells were calculated to estimate the degree of apoptosis (except in the lower left quadrant). In HT1080 cells (A549 cells), they were 91.46% (78.45%) for cells treated with DOX, 66.57% (63.66%) for cells treated with MCP-D, and 91.01% (97.07%) for cells with MSP-D (Figure 6B). In both cells, a significant increase in the degree of cell death was found for the group treated with MSP-D.

Terminal deoxynucleotidyl transferase (TdT) dUTP Nick-End Labeling (TUNEL) assay was adopted to determine cell death-associated DNA fragmentation by CLSM (Figure 6C). Green color was observed in cells treated with free DOX and MSP-D owing to TUNEL staining of apoptotic bodies, the characteristic of apoptosis. Increased TUNEL green fluorescence was clearly observed in cells exposed to MCP-D and MSP-D upon UV light irradiation. The release of DNA small fragments from damaged nuclear envelope and plasma membrane leads to necrotic dispersed TUNEL-positive signals [35]. The mean fluorescence intensity (MFI) of positive cells was analyzed using ImageJ software (Figure 6D). Compared with cells alone, UV-light-treated cells had significantly high percentages of cell death (*p* < 0.0005). MSP-D showed the highest cell-killing effect due to the largest amount of DOX release (Figure 6A). Similar findings were seen in the use of MTT assay (Figure 6E). Taken together, the results implied an enhanced anti-tumor activity by the largest amount of DOX released from MSP-D owing to dual-stimuli response to UV light and GSH.

The cell-killing action of DOX utilized several regulated cell death (RCD) pathways. According to the cell death mechanism revealed by Annexin V-PI dual staining and TUNEL staining, it seems that DOX activated the necroptosis pathway of cell death. Thus, the expression levels of necroptosis-related proteins were assayed. After taken up by HT1080 cells, DOX phosphorylated receptor-interacting serine/threonine-protein kinase 1 (RIP-1) recruited and phosphorylates RIP-3, followed by forming the necroptosome through cellular signaling pathways [30]. The formed necroptosome subsequently phosphorylated the mixed lineage kinase domain-like protein (MLKL), which ruptured the plasma membrane and subsequently oligomerized and inserted into the cell membrane, resulting in hole formation (Figure 2C). This situation caused the loss of membrane potential and integrity and eventually led to cell necroptosis. Compared with the control group, cells exposed to MCP-D and MSP-D upon UV light irradiation displayed higher expression levels of RIP1 and MLKL, indicating the involvement of cell death in necroptosis (Figure 7).

## 4. Conclusions

Novel mPEG-SS-pONBMA diblock copolymers with photo- and redox-responsive properties were successfully synthesized using ATRP. The copolymers could be formulated into micelles with extremely low CMC values. The stimuli-responsive degradability of the copolymers was confirmed according to UV–visible spectra and SEC profiles. Moreover, the degradability of the formulated micelles was observed by DLS and TEM. A clear aldehyde moiety of the copolymer was found in NMR spectra upon UV light irradiation. The micelle alone showed low cytotoxicity against HT1080 cells and A549 cells. Compared with MCP-D, MSP-D showed a synergistic DOX-released profile when treated with GSH and UV light. This dual-stimuli character enhanced the fast disintegration of the micelle in the tumor microenvironment, leading to the highest amount of DOX released from MSP-D and resulting in the highest degree of cell death upon UV light irradiation. The cell death was mainly via necroptosis. This artificially designed micelle is a potential drug delivery carrier.

## Data Availability

Not applicable.

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
