# Peer review of "Light- and Redox-Responsive Block Copolymers of mPEG-SS-ONBMA as a Smart Drug Delivery Carrier for Cancer Therapy"

_pharmaceutics, 2022, doi:10.3390/pharmaceutics14122594_

Round 1

Reviewer 1 Report

The manuscript is well written and the experiments are designed in an organized manner, however, some issues need to be justified.

-In the abstract, it needs revision as some grammatical mistakes are found and some abbreviations are mentioned without the full name. Please add them.

-Please insure that the abbreviations are mentioned in full for the first time in the introduction.

- Please justify why CMC was determined by pyrene and DLS methods?

- Why the cytotoxicity of the prepared micelles was not examined on normal cell line?

- In the 2 examined cell lines, one of them was more senestive  to the prepared micelles, please discuss this and explain why this might happen?

- It will be better to change the IC50 from µg/mL to µM.

- Some experiments in the methodology section lack references.

Reviewer 2 Report

The research paper titled: “Light- and Redox-Responsive Block Copolymers of mPEG-SS-ONBMA as a Smart Drug Delivery Carrier for Cancer Therapy”, study the development of stimuli-responsive polymeric micelles for targeted drug delivery in improving their therapeutic outcome. The manuscript studies the design of copolymers responsive to UV light and GSH.   I recommend the acceptance of the manuscript after major revision.   Comments

1.     There are a lot typographical errors which must be corrected, please revise the manuscript carefully.

2.     The English language of the manuscript needs major revision.

3.     The full name of all abbreviations much be written the first time it is mentioned in the text. Example: GSH, DD water, CLSM.

4.     Experimental part: DCC urea is not the write abbreviation. In addition, the full name must be written.

5.     The procedure in the experimental part needs major revision. Some sentences are too simple and obvious to be written in a research paper, for example “using a Buchner funnel with filter paper” .

6.     The interpretation of the NMR spectra in the Experimental part must be revised and corrected. Especially the multiplicity of all peaks. The author wrote the expected multiplicity and not the one observed. Some of them are surely multiplet peaks.

7.     “2.6. DOX-encapsulated Micelles and DOX Release” part must be rewritten. The procedure is confusing.

8.     “while mPEG-SS-Br exhibited increased intensity of the peak at 1736 cm-1, implying successful esterification.” This hypothesis is not correct. Please correct or remove. First the wave number increase is not significant. In addition, the intensity of the peak is not increased. The interpretation of the IR spectra should be added in the experimental part similar to the H-NMR. In addition, in the caption of IR the Author should mention how the IR spectra was analyzed, for example (whether KBr pellet).

9.     The Authors determined DOX content using a fluorescence spectrometer, which is less accurate than HPLC. The DOX content during encapsulation and release should be performed by HPLC.

10.  Scheme 1 must be redrawn. There is no need to write all the abbreviations in the Scheme. This should be added in the text. The reaction conditions for the preparation of compounds 2 and 4 should be above the arrow. The text in the Scheme is difficult to read.

11.  Scheme 2 must be modified: the name and structure of some compounds are too small. It is difficult to read the text.

12.  Scheme 2 A doesn’t show the proposed degradation route of (MCP) upon UV-light. Scheme 2A should be modified to show the effect of both UV and GSH.

13.   In 3.4. Drug Encapsulation and Cytotoxicity: please correct: “DOX was an anthracycline antibiotic,” to “DOX is an anthracycline antibiotic,”

14.  Part of Figure 4 is hidden. This must be corrected.

15.  Please indicate the IC50 of MCP50, MSP50, free DOX and DOX-loaded MCP50 (MCP-D) and DOX-loaded MSP50 (MSP-D) micelles, in a Figure.

16.  The Caption of Figure 6 is not in accordance with the Figure.

Reviewer 3 Report

Wang et al. described the formulation and characterization of light and redox-sensitive micelles as smart drug delivery nanocarriers. Results described in this manuscript are of interest. However, the authors have to add in the “Materials paragraph” a full description of all the techniques and materials used (see comments below), and several results have to be deeper commented (see comments below).

In view of these general comments and the ones given below, I do recommend the publication of this manuscript in Pharmaceutics after major revision.

Please find below, the specific remarks and/or questions that need to be addressed before any publication.

1. Page 1, Abstract: The abbreviation “GSH” and “mPEG” have to be defined.

2. Page 2, Materials and Methods: One important characteristic of the synthesized polymers is missing. Indeed, (co)polymers have to be analyzed by size exclusion chromatography (SEC). Why did not the authors analyze the synthesized (co)polymers by SEC? The weight (or number) average molar mass (Mw or Mn) and dispersity (Ð) have to be measured and given. Moreover, the conditions used to realized such SEC measurements have to be given: apparatus type, supplier, flow rate, solvent, temperature, standards, detector, etc..

In fact, the authors gave some SEC analysis results (Table 1) without giving all the important and necessary details in the Experimental part.

3. Page 3, paragraph 2.2:

- lines 115 and 118: “MPEG-COOH” and “MPEG-SS-Br” have to be changed to “mPEG-COOH” and “mPEG-SS-Br”

- line 115, “mPEG”: the authors have to give the nature of mPEG end-chains groups. Moreover, mPEG-COOH is commercially available (IRIS Biotech, for example). Why do the authors synthesize such carboxylic acid terminated PEG?

- line 116, “The final product was collected …”: How was collected the final product? What about its purification, its average molar mass and dispersity measurements?

4. Page 3, line 139: The abbreviation “DD” has to be defined.

5. Pages 3 and 4, paragraph 2.4: How do the authors characterize their (co)polymers? It has to be described in the Experimental part.

6. Page 4, line 163: What is the volume of DD water used? It is an important parameter which has an influence on micelles diameter and dispersity, and it has to be specified.

7. Page 4, lines 167-170: What are the conditions used to realized TEM experiments, DLS and CMC measurements? As for SEC analysis, the conditions used for TEM, DLS and CMC measurements must be given.

8. Page 6, line 245: What are the weight (or number) average molar mass and dispersity of the mPEG-COOH?

9. Page 6, line 265 “GPC”: GPC has to be changed to “SEC”, and its meaning has to be given. As said earlier, conditions used to realize the SEC measurements have to be specified.

10. Page 6, line 266: The term “weight” is incorrect. It has to be changed to “molar mass”.

11. Page 7, Figure 1B: What are the peaks between 0.4 and 2 ppm? Are they corresponding to DCU, as I guess? If yes, how can DCU be fully removed?

12.Page 8, Figure 1C: What are the small peaks between 20 to 22 minutes of elution time?

13. Page 8, Table 1: The units (g/mol) have to be given for Mn. Moreover, the terms “polydispersity index (PDI)” and “gel permeation chromatography (GPC)” have to be changed to “dispersity (Ð)” and “size exclusion chromatography (SEC)”.

How do the authors measure the CMC values by DLS? It has to be shortly explained.

14. Page 8, line 306: The nanoprecipitation method has been described by Fessi et al., therefore corresponding references have to be added.

15. Page 9, lines 314 and 315: I suggest to the authors to add a sentence specifying that they chose to continue their work with the MSP50 and MCP50 micelles because of their responsiveness to UV light.

16. Page 9, lines 339 to 340: References have to be added.

17. Page 11, lines 377 to 379: I don’t agree. The SEC chromatograms (It misses the vertical axe for both SEC chromatogram on Figure 3A and 3B), for both micelles before and after GSH and UV light treatments are very similar, excepted for micelles only treated with GSH. If no micelles degradation occurred, what is expected with MCP50 ones, no change should have been observed upon treatment. How can the authors explain the degradation of MCP50 micelles? Moreover, I am not convinced by the TEM images.

18. Page 12: To my opinion, the explanations given in this paragraph are not really clear, and micelles behaviors upon GSH and UV light treatments observed by SEC, DLS and TEM are not consistent. Indeed, if no degradation occurred for MCP micelles upon GSH and UV light treatment, the SEC chromatograms have to be similar, DLS traces must not change and TEM images have to be similar.

19. Page 14, Figure 4: Where are the numbers 1 to 4? It is impossible to fully see the second graph.

Round 2

Reviewer 2 Report

The research paper titled: “Light- and Redox-Responsive Block Copolymers of mPEG-SS-ONBMA as a Smart Drug Delivery Carrier for Cancer Therapy”, study the development of stimuli-responsive polymeric micelles for targeted drug delivery in improving their therapeutic outcome. The manuscript studies the design of copolymers responsive to UV light and GSH.   I recommend the rejection of the manuscript.   I don’t think the manuscript can be accepted for publications in Pharmaceutics. Still there are a lot of errors and mistakes. I tried to revise the corrections, although it is even difficult to revise the manuscript in the presented form.

Reviewer 3 Report

The authors have corrected their manuscript and correctly answered to most of the questions.

Therefore, the manuscript can be published in Pharmaceutics